# Sensitive detection of a bacterial pathogen using allosteric probe-initiated catalysis and CRISPR-Cas13a amplification reaction

Jinjin Shen[1,2], Xiaoming Zhou[3], Yuanyue Shan[1,2], Huahua Yue[1,2], Ru Huang[1,2], Jiaming Hu [1,2]* & Da Xing[1,2]*

The ability to detect low numbers of microbial cells in food and clinical samples is highly valuable but remains a challenge. Here we present a detection system (called 'APC-Cas') that can detect very low numbers of a bacterial pathogen without isolation, using a three-stage amplification to generate powerful fluorescence signals. APC-Cas involves a combination of nucleic acid-based allosteric probes and CRISPR-Cas13a components. It can selectively and sensitively quantify *Salmonella* Enteritidis cells (from 1 to $10^5$ CFU) in various types of samples such as milk, showing similar or higher sensitivity and accuracy compared with conventional real-time PCR. Furthermore, APC-Cas can identify low numbers of *S*. Enteritidis cells in mouse serum, distinguishing mice with early- and late-stage infection from uninfected mice. Our method may have potential clinical applications for early diagnosis of pathogens.

[1] MOE Key Laboratory of Laser Life Science & Institute of Laser Life Science, South China Normal University, Guangzhou 510631, China. [2] College of Biophotonics, South China Normal University, Guangzhou 510631, China. [3] School of Life Sciences, South China Normal University, Guangzhou 510631, China. *email: jmhu@m.scnu.edu.cn; xingda@scnu.edu.cn

Rapid and sensitive detection of pathogenic bacteria from food samples or body fluids is significant since pathogenic bacteria have become a great threat to human health[1]. The current clinical techniques, such as plate counting and polymerase chain reaction (PCR)[2,3], rely on sample enrichment via bacteria culture or extracting total nucleic acids by breaking up a large number of pathogens prior to analysis, which is time-consuming, laborious and expensive. Especially in early stage of pathogen infection, efficient quantification of pathogen-associated nucleic acids with low concentration levels remains a challenge. Over the years, various antibody-based colorimetric[4,5], fluorescence[6–8], electrochemical[9,10] and electrochemiluminescence immunoassays[11] have been developed to address the issues of conventional methods by directly recognizing bacterial cells[12,13]. For example, Abbas et al.[14] achieved attomolar detection of IgG by using cysteine-loaded nanoliposomes in a plasmonic enzyme-linked immunosorbent assay (ELISA) and by inducing the aggregation of gold nanoparticles using hydrolytic agent. Although it achieved remarkable detection limits, this method is a binary (all-or-none) response immunoassay, which is restricted by the stability of liposomes in solution and lack of quantification capability that limits its application in distinguishing early and late stage of pathogen infection. In 2018, Tan et al.[15] used carbon dots (CDs)-encapsulated nanocapsules in a similar sandwich immunoassay setting to release the CDs from nanocapsules, thus generating a fluorescence signal in solution. The detection limit was improved to the single cell level; however, the linear range of this method is extremely narrow (1–200 CFU mL$^{-1}$). In addition, those immunoassays require antibodies, tedious works (nanoparticle synthesis, antibody conjugation etc.) and multiple washing steps, which limit their widespread use in practical application. Therefore, it is vital to develop facile and inexpensive techniques that can capture and identify individual pathogen with high sensitivity and specificity.

A strategy introduced by Winfree et al.[16–18] employing an allosteric DNA molecule by designing functional domains of oligonucleotide sequence that act as a binding unit can catalyze DNA reaction generating a linear signal amplification. In this work, we utilize a similar allosteric probe (AP) for direct identification of bacterial cells. Our AP consists of three functional domains: aptamer domain, primer domain and T7 promoter domain, whose allosteric transformation can be catalytically triggered by identification of target pathogen, and followed by DNA sequence extension and RNA transcription, generating numerous single-stranded RNAs. CRISPR-Cas system has been widely harnessed to gene editing and expression regulation[19–22]. CRISPR-Cas13a, the new member of CRISPR-Cas family, including Cas13a protein (former C2c2) and CRISPR RNA (crRNA), was described to employ crRNA guided Cas13a endonucleases (Cas13a/crRNA) for RNA targeting[23–26]. Recent in vitro study showed that Cas13a/crRNA can be activated to non-specifically cleave nearby RNAs upon specifically recognition of its target RNA (named trans- or collateral cleavage effect), and one activated Cas13a could induce thousands of non-specific RNA degradation[27,28]. However, to the best of our knowledge, there has been no report on allosteric DNA molecules induced CRISPR-Cas13a to quantify pathogens for high signal gain.

Here, we present a combination of allosteric controlled catalysis and collateral cleavage of CRISPR-Cas13a to give a fast, ultrasensitive, DNA-extraction free strategy for direct pathogen detection, termed the "allosteric probe-initiated catalysis and CRISPR-Cas13a" (APC-Cas) system. Twenty percent of world poultry products are contaminated with *Salmonella*, and *Salmonella enterica* serotype Enteritidis (*S.* Enteritidis) is one of the most common *Salmonella* serotypes worldwide[29–31]. Thus, we select *S.* Enteritidis as a model pathogen to verify our assay, in comparison with real-time PCR. Our findings suggest that APC-Cas may potentially be used for pathogenic bacteria detection in food testing and clinical diagnostics.

## Results

**Mechanism of the APC-Cas.** Figure 1 shows an overall illustration of APC-Cas and how it works. A specific allosteric probe (AP), which is a single-stranded DNA molecule, consists of three functional domains: aptamer domain for recognition of target pathogen (purple), primer binding site domain (blue) and T7 promoter domain (yellow). In addition, a phosphate group was labeled at the 3′ end of AP to avoid self-extension and render the DNA molecule resistant to enzymatic hydrolysis[32]. In the absence of target pathogen, the AP is in its non-active configuration with a hairpin structure, and the primer binding site domain and T7 promoter domain are annealed and blocked for the downstream reaction. In the presence of target pathogen, the aptamer domain of AP can specifically recognize and bind with target pathogen, consequently, the hairpin structure of AP will unfold and switch to its active configuration, which allows primers to anneal to the released primer binding site domain. Then, under the participation of DNA polymerase, the AP acts as a template to yield a double-stranded DNA (dsDNA), followed by the displacement of target pathogen for the next catalytic cycle (primary amplification), due to the polymerase extension reaction. Subsequently, T7 RNA polymerase is used to identify the T7 promoter sequence on the generated dsDNA and conduct amplification through transcription process to generate numerous single-stranded RNAs (ssRNAs) (secondary amplification). Finally, in the downstream reaction with Cas13a/crRNA, the crRNA is designed to contain two regions: a guide sequence (28 nt) and a repeat sequence (31 nt). The repeat sequence is an essential part for crRNA to anchor Cas13a enzyme and the guide sequence is complementary to the transcribed ssRNA. When the above transcribed ssRNAs hybridize with Cas13a/crRNA, the collateral cleavage ability of Cas13a/crRNA will be activated to cleave multiple RNA reporter probe (tertiary amplification), thereby generating amplified fluorescence signals.

**Analysis of APC-Cas.** To verify and intuitively display that the AP could specifically bind to the target *S.* Enteritidis and switch to its active configuration, AP is synthesized with a fluorescent moiety covalently linked to the 3′ end of the AP, and a quenching moiety covalently and internally linked to the 5′ end of the stem, as depicted in Fig. 2a. The DNA sequence of aptamer domain in AP was evolved by Kolovskaya et al. against intact *S.* Enteritidis[30]. We chose Clone SE-3 out of the aptamer pool SE7 (23 DNA sequences for *S.* Enteritidis) to be used in our APC-Cas system based on its sequence length (60nt), dissociation constant ($K_d = 7.8 \pm 6.1$ nM), bacteria growth suppression and other performances. We hypothesize that, when the aptamer domain of AP recognizes and binds to the target *S.* Enteritidis, the fluorescent moiety is separated from the quenching moiety and generates fluorescence signal. The laser scanning confocal microscope (LSCM) images in Fig. 2a and Supplementary Fig. 1 confirm that AP successfully binded with *S.* Enteritidis and switched to its active configuration. To further demonstrate the AP-initiated catalysis and isothermal amplification process, the 10% polyacrylamide gel electrophoresis (PAGE) analysis was performed. As shown in Fig. 2b, band in lane 1 (L1) is AP. The AP failed to unfold when only adding primer strand (L2). Even under the addition of Klenow Fragment (KF) DNA polymerase, AP still could not unfold (L3), indicating that the AP has sufficient stability to resist external disturbances. However, when target *S.* Enteritidis was introduced to unfold AP, the primer could hybridize with the unfolded AP and the extension reaction was

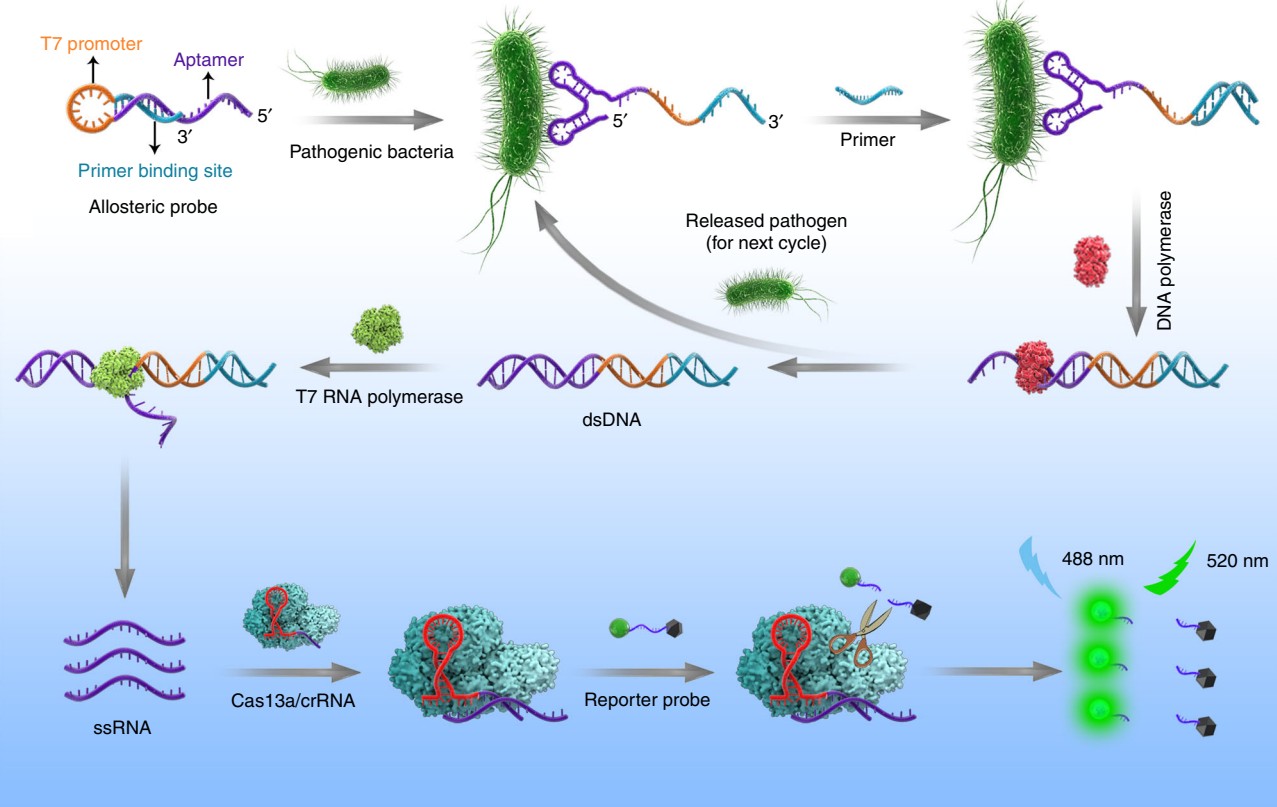

**Fig. 1** Principle of allosteric probe-initiated catalysis and CRISPR-Cas13a (APC-Cas) system for pathogen detection.

initiated by KF DNA polymerase to generate dsDNA with higher molecular weight (L4), which were subsequently recognized by T7 RNA polymerase to transcribe ssRNAs (L5). The collateral cleavage activity of CRISPR-Cas13a used in the downstream reaction was also verified before running the APC-Cas. As shown in Fig. 2c, Cas13a shows perfect activity under the guidance of crRNA in the presence of target RNA strands.

The AP was designed to switch to its active state under the participation of target *S*. Enteritidis, so it must maintain enough stability to resist the interference of target-unrelated primers. In the hairpin structure of AP, the stem length is the key factor to affect its stability. A series of APs with increased stem-lengths (8, 9, 10, 11, 13, 14, and 16 bp) were designed and their functional domains and stems were colored and underlined, respectively, as shown in Supplementary Table 1. The real-time fluorescence signal of these seven APs was monitored, and the result was represented by fluorescence growth rate $V_g$ ($V_g = \Delta F/\Delta t$) (Fig. 2d). Wherein the $\Delta F$ represents the change of the fluorescence intensity before reaching plateau and $\Delta t$ is the time frame of 20 min. Comparing with measuring fluorescence intensity, utilizing the $V_g$ as the parameter of sensing is more time-saving. The background level (without *S*. Enteritidis) with 8-stem AP was 6.4-fold higher than that with 13-stem AP, whereas the signal levels (with *S*. Enteritidis) differed by only 1.3-fold. This large decrease in background and only modest decrease in signal gave the expected increase in signal-to-background ratio (S/BG) when increasing the stem length of AP from 8 bp to 13 bp. The S/BG curve (black line), calculated from each AP, presents the relationship between S/BG and the stability of APs with varied stem lengths. S/BG increases proportionally with the stem length from 8 bp to 13 bp and then decreases from 13 bp to 16 bp. However, the S/BG parameter is not useful if AP is too stable and there is insufficient signal present with respect to the background of measurement, and it is likely that the amount of either signal

or background (or both) using 16-stem AP would be negligible. These are consistent with the results of Gibbs free energy ($\Delta G$) and melting temperature ($T_m$) values (Supplementary Table 2). These results indicated that the AP with a stem length of 13 bp was desirable for the kinetics of state switching and effectively resisting interference of target-unrelated primers, which was selected as the allosteric molecule in our APC-Cas system.

Furthermore, the experimental parameters were optimized to improve the assay performance. First, we investigated the concentration of AP, which acts as catalytic effector and plays an important role in APC-Cas. The signal-to-background ratio increased with the increasing concentration of AP and dropped at a higher concentration of AP (Supplementary Fig. 2a). The reaction rate of catalysis was supposed to be greatly enhanced by increasing the concentration of AP. Furthermore, this would lead to generate a large amount of ssRNA to activate a large quantity of Cas13a, which can improve sensitivity. However, a high concentration of AP might form dimers, which would hinder primer binding and decrease the amplification efficiency. So, AP with the concentration of 400 nM was selected for the following research. Additionally, the concentrations of KF DNA polymerase and T7 RNA polymerase were investigated, respectively. As shown in Supplementary Fig. 2b, c, the optimal concentration of KF DNA polymerase and T7 RNA polymerase were 0.08 U μL$^{-1}$ and 1 U μL$^{-1}$, respectively. Meanwhile, the incubation time of RNA transcription was also optimized. When the incubation time was 60 min, it would obtain the highest signal-to-background ratio (Supplementary Fig. 2d).

**APC-Cas for sensitive *S*. Enteritidis detection**. To verify the sensitivity of APC-Cas, 1, 3 and 8 *S*. Enteritidis cells were picked out and performed with APC-Cas, individually. The results revealed an elevated fluorescence intensity with the increasing

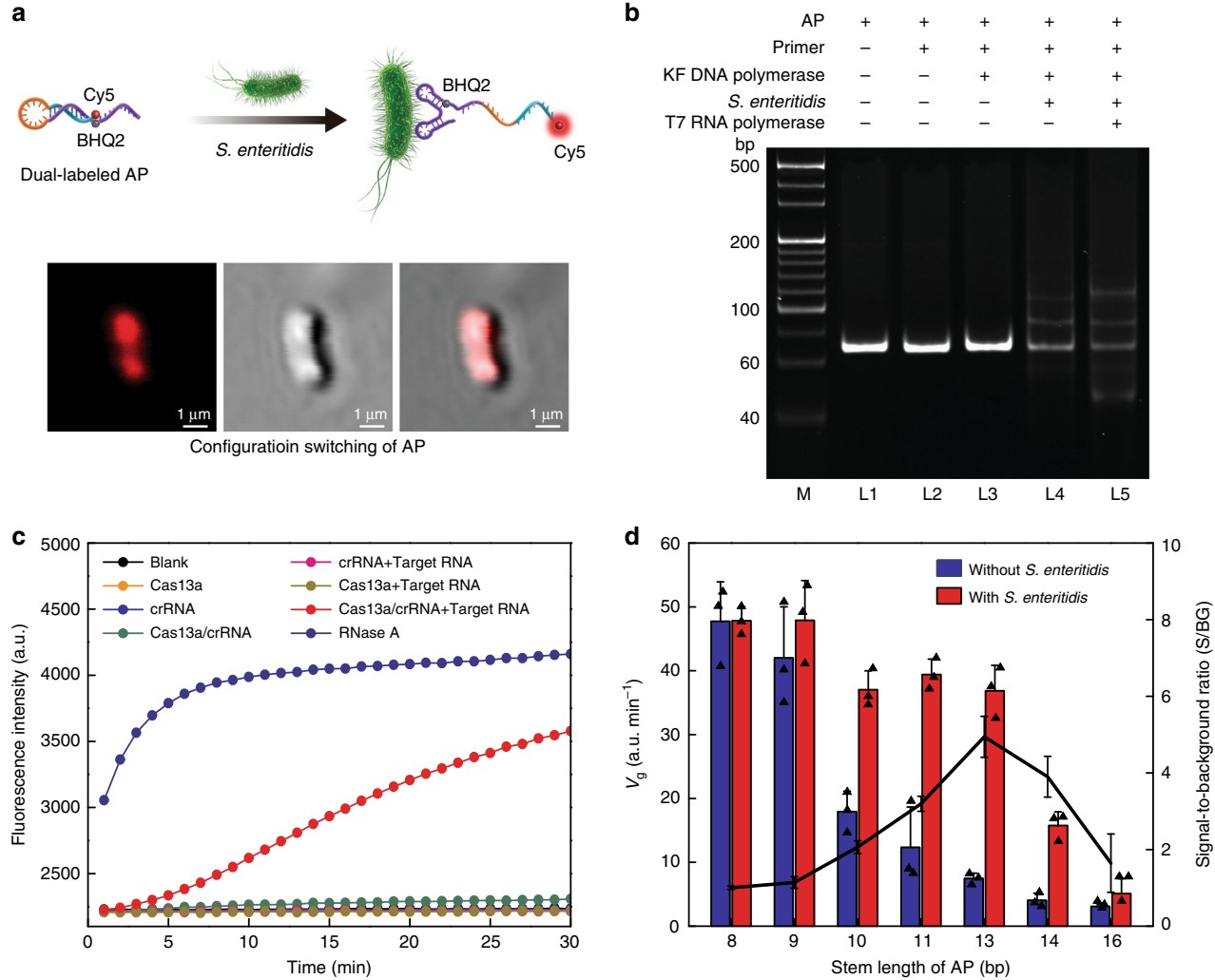

**Fig. 2 Analysis of APC-Cas. a** Illustration and representative laser scanning confocal microscope (LSCM) images of dual-labeled AP binding to the *S. Enteritidis*. **b** Electrophoretic analysis of the feasibility of APC-Cas for *S. Enteritidis* detection. M: DNA marker, L1: AP; L2: AP + primer; L3: AP + primer + KF DNA polymerase; L4: AP + primer + KF DNA polymerase + *S. Enteritidis*; L5: AP + primer + KF DNA polymerase + *S. Enteritidis* + T7 RNA polymerase. '+' means presence, '-' means absence. **c** Fluorescence measurement of LbuCas13a activity. RNase A was used as positive control for the degradation of RNA reporter probe. **d** Comparison of seven APs with varied stem-length. Data represent mean ± s.d., $n = 3$, three technical replicates.

amount of *S. Enteritidis*, and an obvious fluorescence signal can be observed even at single-level (Fig. 3a). The results in Fig. 3b show that APC-Cas can detect a single *S. Enteritidis* cell, due to over 3-fold standard deviations of control (black dash line). The amount of *S. Enteritidis* used in this test was further quantified and demonstrated by the plate colony counting method (Supplementary Fig. 3).

Furthermore, to obtain the dynamic range of our method, a serials of samples containing *S. Enteritidis* cells were prepared under the same condition and detected by APC-Cas simultaneously. Figure 3c shows the background with no *S. Enteritidis* (black trace) and typical fluorescence intensity curves in the presence of varying *S. Enteritidis* levels ($1 - 1 \times 10^5$ CFU). The system was then calibrated versus amounts of *S. Enteritidis*, with responses of fluorescence growth rate ($V_g$) (Fig. 3d). Using direct fluorescent readout at 37 °C, APC-Cas was capable of easily detecting *S. Enteritidis* levels as low as 1 CFU, with a dynamic range extending to $1 \times 10^5$ CFU.

We also compared our APC-Cas system with real-time PCR for the detection of *S. Enteritidis*, since real-time PCR is a widely used method for pathogen detection due to its high sensitivity. real-time PCR reaction was performed using the extracted genomic DNAs from *S. Enteritidis* in the range of 40 to $4 \times 10^7$ CFU (the range of genomic DNAs: $0.1–10^5$ pg μL$^{-1}$) (Fig. 3e, Supplementary Fig. 4). When the amount of *S. Enteritidis* was below 400 CFU, the $C_t$ value was over 30 (Fig. 3f). Even after 40 cycles, there was still no fluorescent peak showing up in the presence of 40 CFU *S. Enteritidis*. The main reason why real-time PCR failed to detect low amount of *S. Enteritidis* may be due to the loss of active components that need to be amplified caused by pathogen disruption and DNA extraction. The result indicates that APC-Cas can be used as a fast and ultrasensitive platform for pathogen detection.

**Specificity of APC-Cas for *S. Enteritidis* detection.** To demonstrate specificity, the APC-Cas was challenged with other bacterial species including *E. coli*, *S. aureus* and *L. monocytogenes*. Figure 4a shows that there was nearly no fluorescence response observed in the presence of $1 \times 10^3$ CFU *L. monocytogenes*, *E. coli*, or *S. aureus*; however, the $V_g$ of *S. Enteritidis* was over 10-fold higher than that of *L. monocytogenes*, *E. coli*, or *S. aureus*. Similar results were obtained when applying APC-Cas for other gram-negative bacteria, such as *Citrobacter freundii*, *Vibrio*

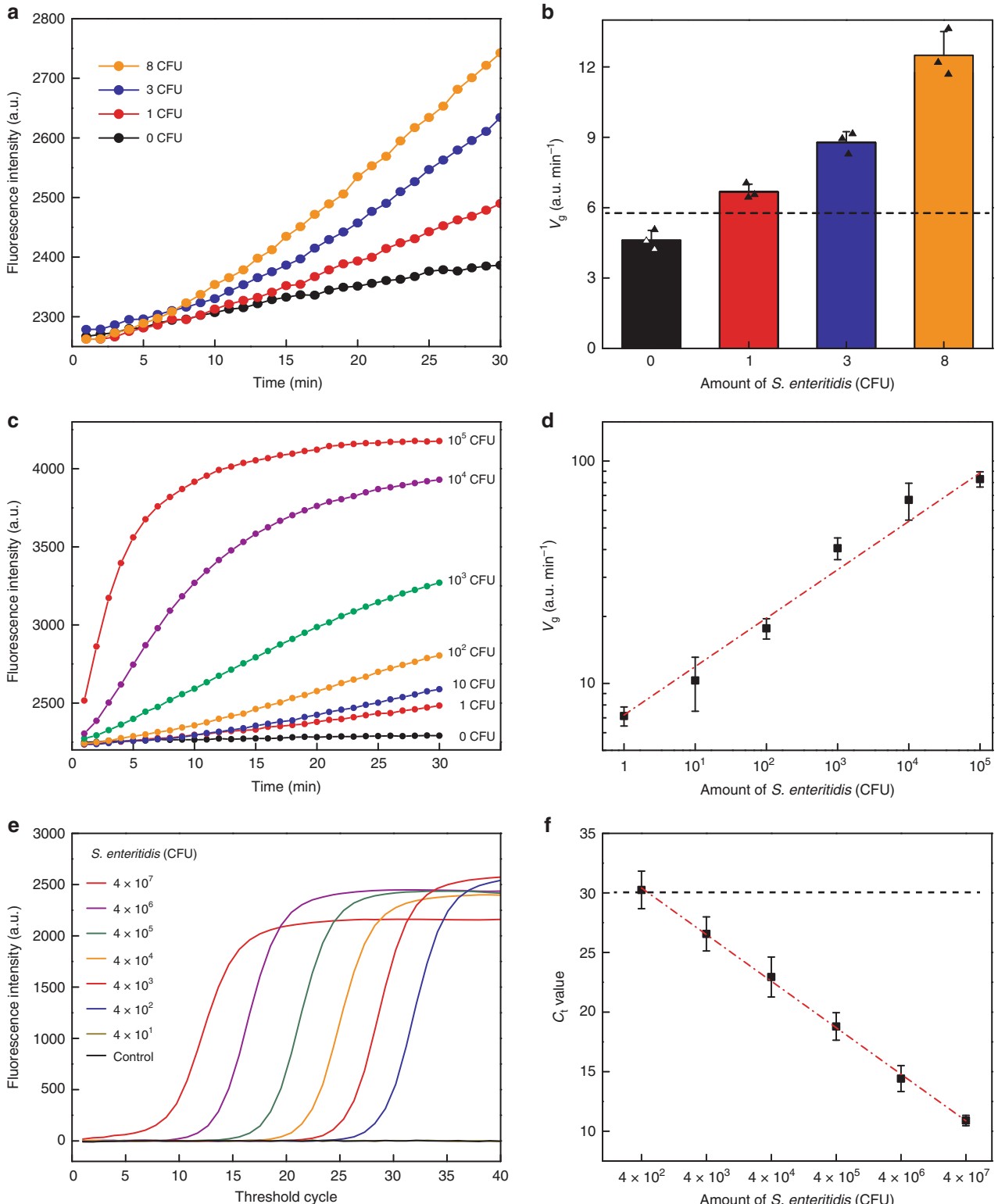

**Fig. 3 APC-Cas for single S. Enteritidis detection. a** The fluorescence signal of APC-Cas for detection of *S.* Enteritidis at single-level. **b** Fluorescence growth rate ($V_g$) of APC-Cas for single *S.* Enteritidis detection. **c** Real-time kinetic measurement of APC-Cas. Different amount of *S.* Enteritidis (0–10⁵ CFU) was introduced into the system at $t = 0$ min. **d** Linear analysis of *S.* Enteritidis detection by APC-Cas. **e** Real-time PCR initiated by amount of *S.* Enteritidis from 40 to $4 \times 10^7$ CFU. **f** Linear analysis of *S.* Enteritidis detection by real-time PCR. Data represent mean ± s.d., $n = 3$, three technical replicates.

*parahaemolyticus*, *Shigella flexneri*, and *Pseudomonas aeruginosa* (Supplementary Fig. 6a).

*S.* Enteritidis is a serovar of *Salmonella enterica* subsp. *enterica*. We showed that APC-Cas could distinguish *S.* Enteritidis from other serovars and subspecies of *S. enterica* (*S. bareilly*, *S. pullorum*, *S. hillingdon*, *S. typhimurium*, and *Salmonella enterica* subsp. *arizonae*) (Supplementary Fig. 6b). Furthermore, we also tested four additional *S.* Enteritidis strains (CMCC 50041, CMCC

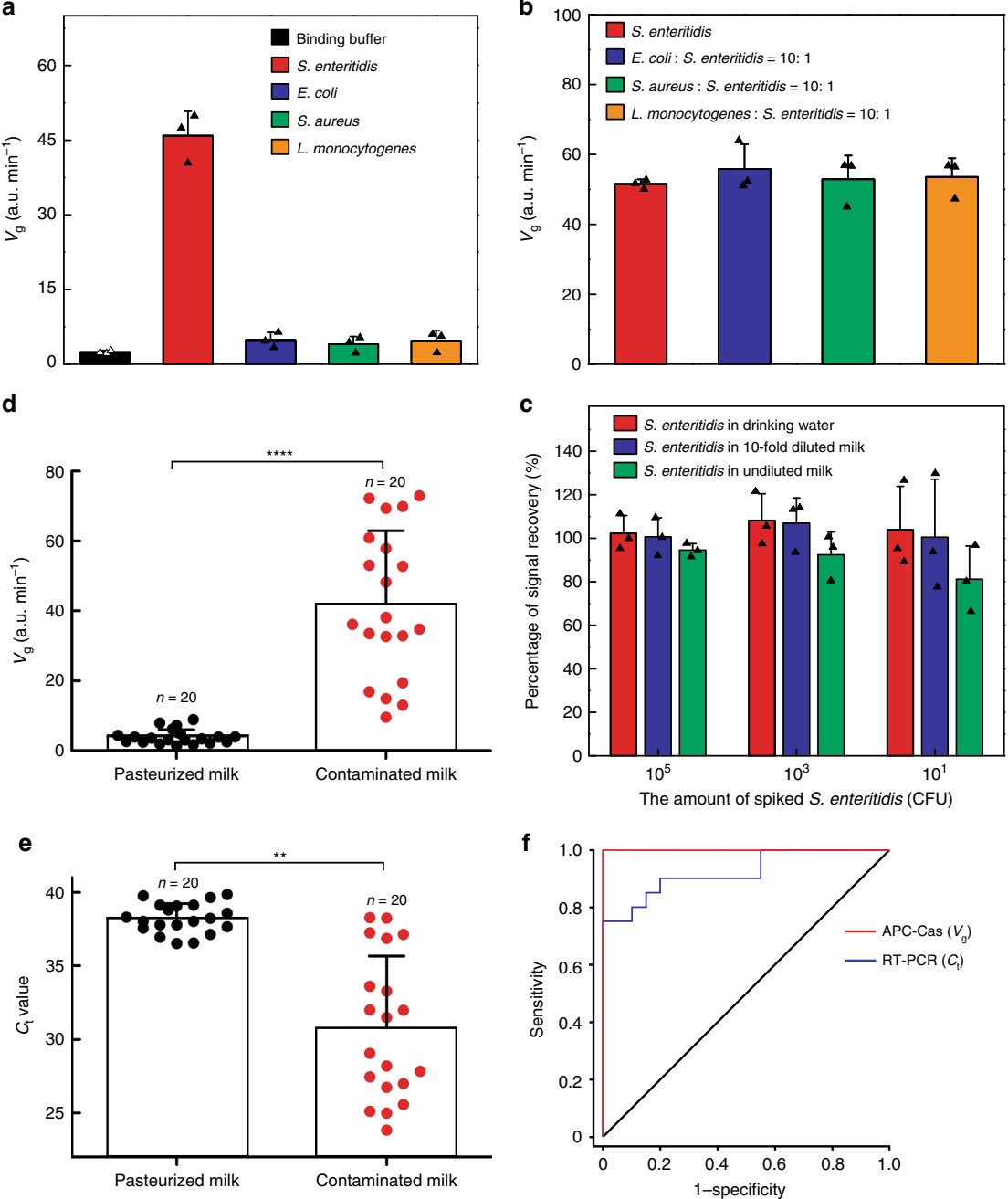

**Fig. 4 Measurement of S. Enteritidis in real sample. a** High specificity was confirmed by challenging the system with three other pathogens (*E. coli*, *S. aureus*, and *L. monocytogenes*). Only *S.* Enteritidis generated measurable responses. **b** The amount of *S.* Enteritidis was one-tenth of other pathogens. **c** Signal recovery of *S.* Enteritidis spiked in drinking water, 10-fold diluted milk and undiluted milk. **d** fluorescence growth rate ($V_g$) of *S.* Enteritidis expressed in the blind validation cohort (paired two-tailed Student's *t* test, ****$P < 0.0001$). **e** $C_t$ value for genomic DNA in *S.* Enteritidis of blind validation cohort (paired two-tailed Student's *t* test, **$P < 0.01$). **f** ROC curve analysis of blind validation cohort. Data represent mean ± s.d., $n = 3$, three technical replicates.

50035, CICC 21527 and CICC 24119) derived from different host species (human and mouse) and sources (Supplementary Fig. 7). The results show that APC-Cas can detect different isolates within the *S.* Enteritidis serotype.

To further investigate the specificity of APC-Cas in complex samples, the test was performed in mixtures of *S.* Enteritidis and other bacteria at 1:10 molar ratio. As expected, there is no significant difference in fluorescence signal among only the *S.* Enteritidis and other complex bacteria samples (Fig. 4b). As shown in Fig. 4c, the signal recovery of APC-Cas was verified

based on spiking 10, $10^3$, $10^5$ CFU *S.* Enteritidis in drinking water, 10-fold diluted milk and undiluted milk, individually. Statistical analysis of fluorescence data revealed that the signal from spiked $10^5$ CFU of *S.* Enteritidis was recovered by 98.5% and 92.5% in 10-fold diluted milk and undiluted milk, respectively, compared with drinking water. Even at spiked 10 CFU of *S.* Enteritidis, the recovery rate was 96.7 and 78.2% in 10-fold diluted milk and undiluted milk, respectively, compared with drinking water. These results are encouraging for the future application of APC-Cas to complex biological samples.

**Measurement of S. Enteritidis in milk and serum.** To further demonstrate the uniqueness of our APC-Cas to detect food samples, we evaluated *S.* Enteritidis levels from pasteurized milk samples ($n = 20$) and contaminated milk samples ($n = 20$) in a blind validation study. APC-Cas was directly applied in these real samples. Fluorescence analysis of blind validation cohorts revealed that the fluorescence signal of the *S.* Enteritidis could effectively distinguish contaminated milk from pasteurized milk ($P < 0.0001$; Fig. 4d). We observed that all 20 contaminated milk samples exhibited higher levels of *S.* Enteritidis than pasteurized milk samples ($P < 0.0001$) (Fig. 4d). Real-time PCR data also revealed a difference in the expression of *S.* Enteritidis genomic DNA between pasteurized milk samples and contaminated milk samples ($P < 0.01$); however, there was still a percentage of signal overlap between these two cohorts (Fig. 4e). The main reason why real-time PCR failed to fully distinguish contaminated milk samples from pasteurized milk samples may be as follows: All bacteria in the sample is lysed together for total DNA extraction regardless of their origins. Especially the DNA targets from low-level *S.* Enteritidis are mixed and highly diluted with DNAs from non-target bacteria. Furthermore, sample preparation process for PCR such as pathogen concentration, DNA extraction and DNA concentration would cause the loss of target pathogens and DNAs. In contrast, our APC-Cas assay does not need pathogen concentration and DNA extraction/concentration. When the AP specifically bind with *S.* Enteritidis, the allosteric switched structure of AP would initiate the catalysis, isothermal amplification and collateral cleavage of CRISPR-Cas13a, directly realizing signal amplification. The ROC curve of APC-Cas showed an AUC of 1.0 in contaminated milk samples compared to pasteurized milk samples, with a sensitivity and specificity of 100% (Fig. 4f; Supplementary Table 3). By contrast, real-time PCR was inferior in classifying contaminated milk samples from pasteurized milk samples (AUC = 0.923), with a sensitivity and specificity of 85% and 85%, respectively (Fig. 4f, Supplementary Table 3).

Finally, we performed APC-Cas to the detection of *S.* Enteritidis in the serum of mice. 8 to 10-week-old mice were infected via oral administration of an *S.* Enteritidis solution, followed by blood collection (Fig. 5a). Uninfected mice were examined as control in parallel. The presence of *S.* Enteritidis in each mouse was evaluated by APC-Cas every 30-min for the first 6 h, and then every 12-h till 5 days (Fig. 5b). It was observed that the mice developed clinical symptoms at 30-min after infection, compared with uninfected mice, and death began at 60-h. For the infected mice model, the $V_g$ increased over time, suggesting *S.* Enteritidis growth in blood. However, a decrease of $V_g$ can be observed from 3 to 5-h, which could be due to partial clearance by the immune system.

The fluorescence analysis of bacteria content in each mouse at indicated time revealed that the $V_g$ of *S.* Enteritidis presence in serum could effectively distinguish infected mice at early stage from healthy mice ($P < 0.001$; Fig. 5c). Also, the *S.* Enteritidis abundance in serum showed an upward trend between infected mice with early stage (2 to 6-h) and late stage (12-h to 5-day). The ROC curve of APC-Cas showed an AUC of 0.997, and a specificity and sensitivity of 100% and 98%, respectively, in each stage of infected mice, supporting its potential for early infection detection (Fig. 5d; Supplementary Table 4).

## Discussion

In this paper, we show that the APC-Cas system that can detect low numbers of *S.* Enteritidis bacterial cells. APC-Cas employs an allosteric probe containing an aptamer domain as identification element for *S.* Enteritidis, a primer binding site domain for

sequence extension and pathogen releasing for catalysis cycle, and a T7 promotor domain for RNA transcription and amplification, combining with collateral cleavage activity of CRISPR-Cas13a system. The signal amplification efficiency of APC-Cas was successfully demonstrated by identifying single cells of *S.* Enteritidis and negligible cross reactions with other bacteria. In addition, the quantification of *S.* Enteritidis in milk samples and serum of infected mice suggest that APC-Cas has potential in food testing and early diagnosis of pathogen infection.

Although APC-Cas is performed in three steps with three types of enzymes and needs to be redesigned and synthesized according to different targets, it does not require bacterial isolation, nucleic acid extraction and washing step, which are needed by other pathogenic bacteria detection methods. Furthermore, the cost of the APC-Cas reagents can be as low as $0.86 per test (Supplementary Table 5), and has other advantages such as short assay time (140 min), high sensitivity and minimal sample preparation requirement (2.5 μL). The system can potentially be used for detection of other microorganisms or macromolecules.

## Methods

**Reagents and materials.** All allosteric probes (APs) and primers were synthesized by Sangon Biotechnology Co. Ltd (Shanghai, China). The RNase inhibitor, 2.5 mM dNTPs, 25 mM MgCl₂, RNase-free water and the dual-labeled (FAM and BHQ1) RNA reporter probe and dual-labeled AP were purchased from Takara Biotechnology Co. Ltd (Dalian, China). Klenow Fragment (3′ → 5′exo⁻), T7 RNA polymerase, ribonucleotide (NTP) solution mix were purchased from New England Biolabs. SYBR green qPCR mix was purchased from Guangzhou Dongsheng Biological Technology Co. Ltd. *Salmonella enteritidis* (CMCC 50040), *Listeria monocytogenes* (ATCC 19115), *Escherichia coli* (O157: H7 GW1.0202), *Staphylococcus aureus* (CMCC 26003), *Citrobacter freundii* (ATCC 43864), *Vibrio parahaemolyticus* (ATCC 17802), *Shigella flexneri* (CMCC 51572), *Pseudomonas aeruginosa* (ATCC 15442), *S. bareilly* (ATCC 9115), *S. pullorum* (ATCC9120), *S. hillingdon* (ATCC 9184), *S. typhimurium* (ATCC 14028) and *S. arizonae* (ATCC 700155) were purchased from the Guangdong Microbial Culture center (Guangzhou, China). *Salmonella enteritidis* (CMCC 50041, CMCC 50035) was purchased from National Center For Medical Culture Collections (Beijing, China). *Salmonella enteritidis* (CICC 21527, CICC 24119) was purchased from China Center of Industrial Culture Collection (Beijing, China). The pET-Sumo-LbuCas13a plasmid was a generous.pngt from Yanli Wang (Institute of Biophysics, Chinese Academy of Sciences, Beijing, China).

**LbuCas13a protein expression and purification.** The LbuCas13a expression and purification were performed as our previous work[33]. Briefly, The *E. coli* Rosetta (DE3) was transformed with pET-Sumo-LbuCas13a expression plasmid and grown overnight in Terrific Broth (TB) medium at 37 °C and 150 rpm until the exponential growth phase. Afterwards, protein expression was induced with 100 μM isopropyl-1-thio-b-D-galactopyranoside (IPTG) and cultured at 16 °C for 12 h. Cells were harvested by centrifugation at 5000 rpm and lysed by sonication in the lysis buffer (20 mM Tris–HCl, 1 M NaCl, 20 mM imidazole, 10% glycerol, pH 7.5). Lysate was separated by centrifugation and the supernatant was incubated with Ni-NTA agarose, the bound protein was eluted by elution buffer (20 mM Tris-HCl, pH 7.5, 150 mM NaCl, 250 mM imidazole). The His₆-Sumo tag of LbuCas13a protein was digested by Ulp1 protease and further purified by heparin column (GE Healthcare). The purified product was dissolved in storage buffer (20 mM Tris–HCl, pH 7.5, 1 M NaCl, 50% glycerol) and stored at −80 °C until use.

**Allosteric probes and crRNA preparation.** The APs (Supplementary Table 1) were dissolved in 1 × NEBuffer 2 (50 mM NaCl, 10 mM Tris-HCl, 10 mM MgCl₂, 1 mM DTT, pH 7.9) and primer (Supplementary Table 1) was dissolved in RNase-free water. Before use, APs solution was incubated at 95 °C for 5 min and following gradient cooled (2 °C min⁻¹) to room temperature to ensure that APs correctly folded into a hairpin structure, then stored at 4 °C for later use.

The crRNA of LbuCas13a was produced by in vitro transcription using T7 RNA polymerase according to the previous design by our team with some modifications[34]. Briefly, double stranded DNA templates containing T7 promoter sequence were prepared by gradient cooling (holding at 95 °C for 5 min and then gradient cooled (5 °C min⁻¹) to room temperature). Then the transcription reaction was performed with DNA templates, T7 RNA polymerase NTP mix and 1 × reaction buffer (40 mM Tris-HCl, 2 mM spermidine, 1 mM dithiothreitol, 11 mM MgCl₂, pH 7.9) at 37 °C for 6 h. The transcription product was purified by RNA clean Kit (Tiangen), then analyzed by polyacrylamide gel electrophoresis (PAGE) (Supplementary Fig. 5), quantified by Nanodrop 2000 (Thermo Fisher) and stored at −80 °C for later use.

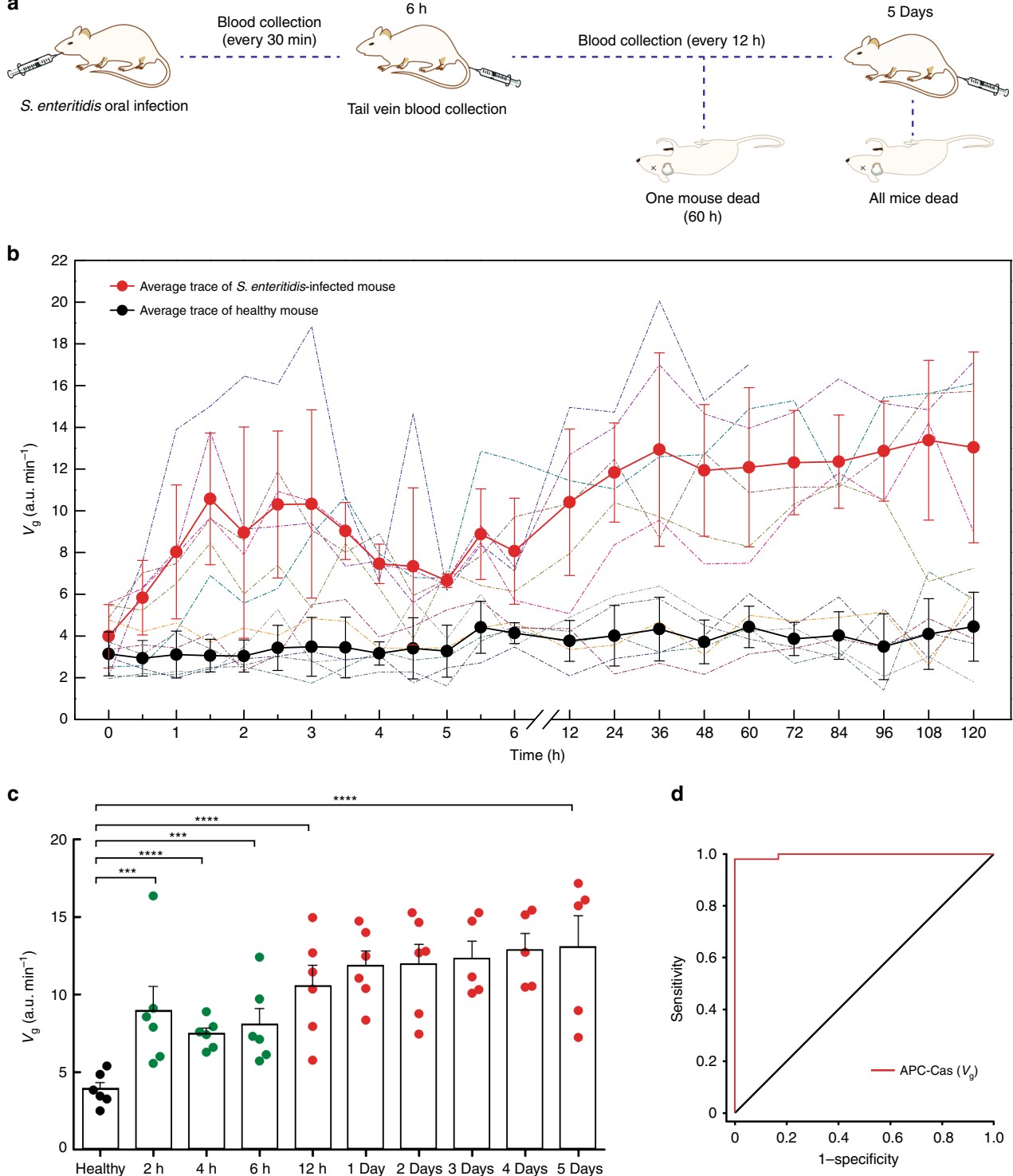

**Fig. 5 Measurement of _S. Enteritidis_ in mouse serum. a** Schematic illustration for the construction of _S. Enteritidis_-infected mouse model and blood collection. **b** _S._ Enteritidis expression curves versus time of six infected mice and six healthy mice (dash-dot lines), with the average trace (red line, black line) shown; error bars represent standard deviations; blood collected from 8 to 10-week-old healthy ($n = 6$) and _S._ Enteritidis-infected ($n = 6$) mice. **c** Fluorescence growth rate ($V_g$) of _S._ Enteritidis expressed in serum of infected mice and healthy mice at indicated time (paired two-tailed Student's _t_ test, ***$P < 0.001$, ****$P < 0.0001$). **d** ROC curve analysis of healthy mice ($n = 6$) versus _S._ Enteritidis-infected mice ($n = 6$) at early stage (2 to 6-h) and late stage (12-h to 5-day). The mice and syringes used in Fig. 5a were the templates of ChenBioDraw Ultra 14.0 software. Bars represent mean ± s.d.

**Bacteria sample preparation**. All the bacteria strains were grown in Luria-Bertani (LB) medium (1 g tryptone, 1 g NaCl, 0.5 g yeast extract, 100 mL sterilized water, pH 7.0) to the exponential growth phase and harvested by centrifugation at 5000 rpm for 5 min. The collected bacteria were resuspended in binding buffer (30 mM MES, 100 mM NaCl, pH 6.0). The concentration of bacteria was assayed by traditional plate-counting method. All materials in

contact with the bacteria were sterilized in an autoclave at 121 °C for 30 min before and after use.

**APC-Cas for _S._ Enteritidis detection**. About 2.5 μL sample was incubated with APs at room temperature for 30 min. Subsequently, the amplification assay was

carried out with reaction mixture containing $1 \times$ NEBuffer 2, 0.08 U $\mu L^{-1}$ Klenow Fragment ($3' \rightarrow 5'$exo$^-$), 200 $\mu$M dNTP and primer at 37 °C for 20 min, the primer concentration is twice that of the AP. After extension, the transcription was carried out with $1 \times$ reaction buffer (40 mM Tris-HCl, 2 mM spermidine, 1 mM dithiothreitol, 11 mM MgCl$_2$, pH 7.9), 5 mM NTP mix, 1 U $\mu L^{-1}$ T7 RNA polymerase and 1 U $\mu L^{-1}$ RNase inhibitor at 37 °C for 60 min.

The fluorescence assay was performed with 10 nM purified LbuCas13a, 10 nM crRNA, 200 nM dual-labeled (FAM and BHQ1) RNA reporter probe and varying amplification product in $1 \times$ reaction buffer (10 mM Tris-HCl, 50 mM KCl, 1.5 mM MgCl$_2$, pH 8.3) at 37 °C for 30 min on CFX real-time PCR detection systems (FAM channel), and fluorescent kinetics were measured every minute.

**Detection of *S*. Enteritidis by real-time PCR.** The *S*. Enteritidis genomic DNA was extracted by Bacterial Genomic DNA Extraction Kit (Tiangen Biochemical Technology (Beijing) Co., Ltd.) and resuspended in 50 $\mu$L of deionized water. After gradient dilutions, the number of *S*. Enteritidis was quantified by plate counting method. Taken the *S*. Enteritidis fimbriae gene A (sefA) as target sequences, primers were designed using the Primer Premier 6.0 software. Then real-time PCR was performed with $1 \times$ SYBR green qPCR mix, 200 nM primer pair and different concentrations of 5 $\mu$L genomic DNA in 25 $\mu$L reaction mixture and carried out on CFX real-time PCR detection systems.

**Detection of *S*. Enteritidis in food samples.** Milk and drinking water were purchased from the local supermarket. For specificity tests, fixed amount of bacteria (*S*. Enteritidis, *E. coli*, *S. aureus* or *L. monocytogenes*, or their mixture) was spiked into milk and drinking water, respectively. For validation test, varied unknown amount of *S*. Enteritidis, in the range of 10 to $1 \times 10^4$ CFU, was randomly spiked into 20 milk samples, and then messed with 20 pasteurized milk samples. These 40 milk samples were prepared by a third party as double-blind preparation, APC-Cas and real-time PCR were used to detect *S*. Enteritidis in the blind validation cohort, respectively.

**Establishment of a mouse infection model.** The C57BL/6 mice were purchased from Guangdong Medical Laboratory Animal Center. All animal studies were conducted in accordance with the guidelines of the National Regulation of China for Care and Use of Laboratory Animals, and were performed with the approval of Scientific Research Ethics Committee of South China Normal University (South China Normal University, Guangzhou, China). The *S*. Enteritidis was grown in LB medium and harvested by centrifugation at 5000 rpm for 5 min. In total 12 8–10-week-old mice were randomly divided into 2 groups, one group was infected through oral with 100 $\mu$L *S*. Enteritidis solution ($10^9$ CFU), another group as a control. During the early stages (0–6 h), the blood samples (20 $\mu$L) of mice were collected every 30 min of infection. After 12 h, blood samples were collected every 12 h up to 120 h. Whole-blood samples were collected into tubes containing heparin and stored at 4 °C for later use. Serum was obtained by centrifugation at 1000 rpm for 3 min.

**Statistical analysis.** All experiments and assays were repeated at least three times. The data were expressed as mean ± s.d. and compared by Student's *t* test. The GraphPad Prism version 5.0, IBM SPSS Statistics version 19.0 and Origin 8 were used for data analysis.

**Reporting summary.** Further information on research design is available in the Nature Research Reporting Summary linked to this article.

## Data availability

The data supporting the findings are available in this manuscript and its Supplementary Information files, or from the corresponding author upon request. Source data underlying figures and supplementary figures are provided as a Source Data file, and are also available from Figshare (https://doi.org/10.6084/m9.figshare.11309750).

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

## Acknowledgements
This work was supported by the National Natural Science Foundation of China (NSFC) (81630046; 91539127; 21475048; 21874049; 21804023), and the Scientific Research Projects of Guangzhou (201805010002). We thank Professor Yanli Wang for providing the pET-Sumo-LbuCas13a expression vectors.

## Author contributions
X.Z., J.H. and D.X. conceived the concept; J.H., D.X. and J.S. designed the experiments; J.S., Y.S., H.Y. and R.H. performed the experiments; J.S. and J.H. analyzed data and wrote the paper; J.H. and D.X. commented and revised the paper.

## Competing interests
The authors declare no competing interests.
