## [Peer Review File · Nature Communications]

Reviewers' comments:

Reviewer #1 (Remarks to the Author):

The study by Shen, et al. describes a new and innovative assay to detect a pathogen, such as *Salmonella enteritidis* contamination in food or body fluids. The assay utilizes an allosteric probe, which is composed of an aptamer sequence specific for the target bacteria (*S. enteritidis*). Although poor signal-to-noise limits the sensitivity of such detection methods, this assay utilizes a CRISPR-Cas13a step to amplify a fluorescence signal. In sum, the assay appears more sensitive and faster than conventional real-time PCR methods, suggesting its potential in applications involving disease diagnosis and food safety inspection.

The approach is innovative, but there are some limitations that might prevent it from becoming widely adopted.

1. There are insufficient details about the aptamer. What is its molecular target? Was it discovered against a particular protein, or against intact *S. enteritidis*? Did the authors discover it, or did someone else? If the latter, please cite accordingly. What are its binding kinetics?
2. The aptamer is specific for only *S. enteritidis*, so expanding the assay for other pathogens would require discovering new aptamers for different targets.
3. The panel of other tested bacteria (*E. coli*, *S. aureus*, and *L. monocytogenes*) is quite narrow. Cross-reactivity might still exist with other strains, particularly for other gram-negative bacteria. Thus, the possibility of 'false positive' reactivity is also uncertain.
4. It is also unclear whether the aptamer and assay is compatible for the wide diversity of *S. enteritidis* subspecies and strains. So, the possibility of 'false negative' reactivity is still unclear.
5. Could the method be applied for antibiotic resistant bacteria? It does not appear so, since the aptamer could not distinguish between resistant and non-resistant strains.

Overall, the assay offers potential improvement over existing approaches, but is quite limited—specifically for only *S. enteritidis*. This study is appropriate for a specialized journal.

Reviewer #2 (Remarks to the Author):

These authors demonstrate novel use of a Cas13 Crisper system to identify pathogenic bacteria without the use of culturing or nucleic acid extraction. They provide convincing examples of the efficacy of this approach in biological samples. The only concerns I have are the evolution of the aptamers which are the selective binding agents. How many different aptamers have been evolved and tested in this assay system? Furthermore are the aptamers stabilized against nucleases by some form of chemical modification?

Reviewer #3 (Remarks to the Author):

Manuscript Review:

Title: Single Pathogen Detection via Allosteric Probe-initiated Catalysis and CRISPR-Cas13a Amplification Reaction"

Authors: Shen et al.

1- The manuscript by Shen et al describes a new method for microbial detection. The method is based on the combination of two known techniques: allosteric DNA probes and CRISPR-Cas enzymes. The Allosteric DNA probes have already been used for detection, and the authors here combine these probes with a CRISPR-Cas system with the purpose of adding two levels of signal amplification through DNA replication and transcription to improve the detection limit. This combination is novel and has not been reported before, and to my knowledge, this is the first use of CRISPR techniques for microbial detection. The manuscript is well-written and technically sound.

2- I have two main concerns:

A- The problem with aptamers: The efficiency of using aptamers for large targets such as microorganisms. Despite the abundant literature in the field, there is no commercial detection assay using aptamers as receptors because of their lack of specificity and poor reproducibility. The only commercial use of aptamers today is for therapeutic reasons and on very small targets (molecules). Given these concerns, the authors should at least provide more information on the aptamers used in this work. Are they commercially obtained, are they produced by the team? did they characterize the binding properties of the aptamers (affinity, dissociation constants)?

B-The problem with "single pathogen" detection: One of the major problems when we attempt to perform low detection limits with low number of cells is that the probability of having microorganisms in your diluted samples decreases exponentially. If we dilute 10 times a 1mL solution with 10 CFU, the result is rarely 1 CFU per mL. The chance to get 1 CFU per mL is very slim. As a result, one should analyze all the diluted samples to ensure a positive detection, and even then, it's hard to demonstrate that there is only one single pathogen your sample. I would suggest avoiding titles with "single pathogen" detection claims.

3- Cost of the assay: One of the main motivations of the authors is to offer a sensitive and affordable microbial detection assay. It is true that the developed assay overcomes the need for antibodies. However, this assay uses three different types of enzymes: DNA polymerase, RNA polymerase and CRISPR-Cas13a, and require equipment for DNA amplification, which would significantly increase the cost of the assay. It would be helpful if the authors could estimate the cost of the assay for a single test.

4- Time of detection: The proposed assay involves 5 biochemical steps: target recognition by the aptamers, DNA replication, RNA transcription, RNA recognition by the enzyme Cas13, and Cleavage of the reporter RNA by Cas13. It would be helpful if the authors estimate the time needed for each step and the overall time of the assay.

We appreciate all reviewers for their valuable comments and suggestions. Our co-authors have conducted a series of experiments to address the raised concerns in the past three months. Our response to review comments together with new experimental results is given in this letter and the revised manuscript (marked in green). In the following, we present our response (marked in blue) to each review comment (marked in yellow) in detail.

Reviewers' comments:

Reviewer #1 (Remarks to the Author):

The study by Shen, et al. describes a new and innovative assay to detect a pathogen, such as *Salmonella enteritidis* contamination in food or body fluids. The assay utilizes an allosteric probe, which is composed of an aptamer sequence specific for the target bacteria (*S. enteritidis*). Although poor signal-to-noise limits the sensitivity of such detection methods, this assay utilizes a CRISPR-Cas13a step to amplify a fluorescence signal. In sum, the assay appears more sensitive and faster than conventional real-time PCR methods, suggesting its potential in applications involving disease diagnosis and food safety inspection.

The approach is innovative, but there are some limitations that might prevent it from becoming widely adopted.

1. There are insufficient details about the aptamer. What is its molecular target? Was it discovered against a particular protein, or against intact *S. enteritidis*? Did the authors discover it, or did someone else? If the latter, please cite accordingly. What are its binding kinetics?

Answer: Thanks for the concern on this issue. The target of aptamer is intact live *S. enteritidis*. The DNA sequence of aptamer domain in allosteric probe used in our paper was Clone SE-3 (60 nt) which was discovered by Kolovskaya et al.¹ against intact *S. enteritidis*. The aptamer sequence marked in purple is shown in **Supplementary Table 1**. The dissociation constant (K_d) of this aptamer is 7.8 ± 6.1 nM. We have added the associated description and following reference on **page 7** in the revised manuscript.

1. Kolovskaya, O.S. et al. Development of Bacteriostatic DNA Aptamers for Salmonella. *Journal of Medicinal Chemistry* **56**, 1564-1572 (2013).

2. The aptamer is specific for only *S. enteritidis*, so expanding the assay for other pathogens would require discovering new aptamers for different targets.

Answer: We agree with the reviewer's comment. The aptamer domain of allosteric probe (AP) designed in this work is specific for only *S. enteritidis*. If expanding the assay for detecting different pathogens, the sequence of aptamer domain in AP should be changed according to the target pathogen.

3. The panel of other tested bacteria (*E. coli*, *S. aureus*, and *L. monocytogenes*) is quite narrow. Cross-reactivity might still exist with other strains, particularly for other gram-negative bacteria. Thus, the possibility of 'false positive' reactivity is also uncertain.

Answer: We thank the reviewer for this suggestion. Four other gram-negative bacteria samples including *Citrobacter freundii* (ATCC 43864), *Vibrio parahaemolyticus* (ATCC 17802), *Shigella flexneri* (CMCC 51572) and *Pseudomonas aeruginosa* (ATCC 15442) were chosen randomly. A specificity experiment among *S. enteritidis* and these four gram-negative bacteria was performed under the same experimental condition using APC-Cas system based on reviewer's suggestion (**Supplementary Fig. 6A**). The results revealed there was nearly no fluorescence response observed in the presence of 1×10^3 CFU *Citrobacter freundii*, *Vibrio parahaemolyticus*, *Shigella flexneri* or *Pseudomonas aeruginosa*; however, the fluorescence growth rate (V_g) of *S. enteritidis* was near 10-fold higher than that of *Citrobacter freundii*, *Vibrio parahaemolyticus*, *Shigella flexneri* or *Pseudomonas aeruginosa*. These results demonstrate no false positive reaction when detecting non-target gram-negative bacteria. We have added **Supplementary Fig. 6A** and associated description on **page 15** in the revised manuscript.

In particular, *S. enteritidis* is one serovar of *Salmonella enterica subsp. Enterica*. To further investigate the possibility of 'false positive' of APC-Cas, four other serovars of *Salmonella enterica subsp. Enterica*, including *S. bareilly* (ATCC 9115), *S. pullorum* (ATCC9120), *S. hillington* (ATCC 9184), *S. typhimurium* (ATCC 14028), and another *Salmonella enterica subsp.*

Arizonae (ATCC 700155), were chosen randomly. The comparison experiments among *S. enteritidis* and these five *Salmonella enterica* subspecies and serovars were performed under the same experimental condition using APC-Cas system (**Supplementary Fig. 6B**). The results show that APC-Cas system could barely detect *S. Bareilly*, *S. pullorum*, *S. hillington*, *S. typhimurium* or *S. arizonae*, but it could identify *S. enteritidis* and generate amplified fluorescence signal. The APC-Cas system shows a good specificity in distinguishing *S. enteritidis* from the wide diversity of *Salmonella enterica* subspecies and serovars. We have added **Supplementary Fig. 6B** and associated description on **page 15** in the revised manuscript.

Supplementary Fig. 6 | Specificity test of APC-Cas system for *S. enteritidis*. (A) Comparison of fluorescence growth rate (V_g) of APC-Cas system among *S. enteritidis* and non-target gram-negative bacteria (*C. freundii*, *V. parahaemolyticus*, *S. flexneri*, *P. aeruginosa*). (B) Comparison of V_g of APC-Cas system among *S. enteritidis* and other *Salmonella enterica* subspecies and serovars, including *S. bareilly*, *S. pullorum*, *S. hillington*, *S. typhimurium* and *S. arizonae*.

4. It is also unclear whether the aptamer and assay is compatible for the wide diversity of *S. enteritidis* subspecies and strains. So, the possibility of 'false negative' reactivity is still unclear.

Answer: Thanks for this concern. Besides the *S. enteritidis* strain (CMCC 50040) used in the manuscript, four additional *S. enteritidis* strains (CMCC 50041, CMCC 50335, CICC 21527 and CICC 24119) derived from different host species (human and mouse) were collected from the diverse institutes, as shown in the **table of Supplementary Fig. 7**. The comparison

experiments among these five *S. enteritidis* strains were performed under the same experimental condition using APC-Cas (**Supplementary Fig. 7**). **Supplementary Fig. 7A** shows all five *S. enteritidis* strains give significant fluorescence increase at the level of 1×10^3 CFU within 30 min, comparing to the background with no *S. enteritidis*. In **Supplementary Fig. 7B**,

Strains	Serovars	Host species	Sources
---------	----------	--------------	---------

The fluorescence growth rate (V_g) of all five *S. enteritidis* strains is over at least 12-fold higher

Supplementary Fig. 7 | Comparison of different *S. enteritidis* strains within the same serotype using APC-Cas. (A) The fluorescence signal of APC-Cas for detection of five *S. enteritidis* strains. (B) Fluorescence growth rate (V_g) of APC-Cas for detection of five *S. enteritidis* strains.

than that of background, and the V_g of those four *S. enteritidis* strains differs from 94% to 130% of the *S. enteritidis* strain (CMCC 50040) used in the manuscript. Based on these results, it seems the aptamer and assay is compatible for different *S. enteritidis* strains within the same serotype. We have added **Supplementary Fig. 7** and associated description on **page 15 and 16** in the revised manuscript.

CMCC 50040	S. enteritidis	Human	Guangzhou Institute of Microbiology, China
CMCC 50041	S. enteritidis	Human	National Center For Medical Culture Collections, China
CMCC 50335	S. enteritidis	Mouse	National Center For Medical Culture Collections, China
CICC 21527	S. enteritidis	Human	China Center of Industrial Culture Collection
CICC 24119	S. enteritidis	Human	China Center of Industrial Culture Collection

5. Could the method be applied for antibiotic resistant bacteria? It does not appear so, since the aptamer could not distinguish between resistant and non-resistant strains.

Answer: We thank the reviewer for this concern. At present, some *S. enteritidis* strains appear to be insensitive to antibiotics in clinical practice. However, these strains carry different drug resistance genes, which leads to inconsistent drug resistance phenotypes. Methicillin-resistant *Staphylococcus aureus* (MRSA), which is typically called as a “superbug” in the *Staphylococcus aureus* (SA) family,² is with a clear drug-resistant phenotype and a clinically common bacteria. So, MRSA was chosen as a model to study whether APC-Cas system could be applied for antibiotic resistant bacteria. We redesigned and synthesized a new allosteric probe with the aptamer domain specifically targeting MRSA. The DNA aptamer against MRSA was developed by Ocsoy et al.³ The DNA sequences used in this experiment is provided below in **Table L1**. SA was used as control. The results in **Fig. L1** showed that APC-Cas could truly distinguish MRSA from SA under the same amount of bacteria (1×10^3 CFU). The V_g of MRSA was over 3.5-fold higher than that of SA. These new results demonstrate that APC-Cas could be applied for antibiotic resistant bacteria detection in the future.

Although these results show the possibility of antibiotic resistant bacteria detection of APC-Cas, we feel that the redesigned assay for testing MRSA is preliminary. Therefore, we decide that these results and associated information for MRSA test are just shown in rebuttal

letter, not in the manuscript.

2. Bartlett, J.G. Antibiotic-Associated Diarrhea. *New England Journal of Medicine* **346**, 334-339 (2002).

3. Ocsoy, I. et al. DNA aptamer functionalized gold nanostructures for molecular recognition and photothermal inactivation of methicillin-Resistant *Staphylococcus aureus*. *Colloids and Surfaces B: Biointerfaces* **159**, 16-22 (2017).

Fig. L1 | Methicillin-resistant *Staphylococcus aureus* (MRSA) detection by redesigned APC-Cas system. (A) The fluorescence signal of redesigned APC-Cas for identifying *MRSA* from *S. aureus*. (B) Fluorescence growth rate (V_g) of redesigned APC-Cas for *MRSA* detection.

Table L1 | Single-stranded DNA sequences used in redesigned APC-Cas for Methicillin-resistant *Staphylococcus aureus* (MRSA)

Name	(abbreviation)	Nucleic acid sequence, listed 5' to 3'
Allosteric probe for MRSA	(AP-MRSA)	ATCCAGACGTGACGCAGCATGCGGTTGGTTGCGGTT GGGCATGATGATTTCTGTGTGGACACGGTGGCTTAG TACCTATAGTGAGTCGTATTAGGTAAGCCA-P
Primer	(Primer)	AGTACCTA
Template strand 1 of MRSA crRNA	(T1-MRSA -crRNA)	GCCCTTAATACGACTCACTATAGGGGACCACCCAAA AATGAAGGGGACTAAAACCATGATGTATTTCTGTGTG GACACGGTG
Template strand 2 of MRSA crRNA	(T2-MRSA-crRNA)	CACCGTGTCCACACAGAAATACATCATGGTTTTAGTC CCCTTCATTTTTGGGGTGGTCCCCTATAGTGAGTCG TATTAAGG GC

The DNA sequence marked in purple colour represents aptamer domain; the DNA sequence marked in yellow colour represents T7 promoter domain; the DNA sequence marked in blue colour represents primer binding site domain.

Overall, the assay offers potential improvement over existing approaches, but is quite limited—specifically for only *S. enteritidis*. This study is appropriate for a specialized journal.

Answer: We hope we have addressed the specificity issue mentioned by the reviewer by applying APC-Cas system for other gram-negative bacteria, *Salmonella enterica* subspecies and serovars, and antibiotic resistant bacteria. These new results demonstrate that the APC-Cas system is extremely flexible, could be redesigned and synthesized for different targets, since any target with aptamers could be assayed via APC-Cas system. Our work is relevant to the area of “Health sciences” and “Biological Sciences”, which is suitable to the journal *Nature Communications*.

Reviewer #2 (Remarks to the Author):

These authors demonstrate novel use of a Cas13 Crispr system to identify pathogenic bacteria without the use of culturing or nucleic acid extraction. They provide convincing examples of the efficacy of this approach in biological samples. The only concerns I have are the evolution of the aptamers which are the selective binding agents. How many different aptamers have been evolved and tested in this assay system? Furthermore are the aptamers stabilized against nucleases by some form of chemical modification?

Answer: We deeply appreciate the positive comments on our work. The DNA sequence of aptamer domain in allosteric probe (AP) used in this assay was evolved by Kolovskaya et al.¹ Kolovskaya et al. has developed a Cell-SELEX technique for selecting highly specific DNA aptamers to *S. enteritidis*. In the best aptamer pool SE7 (23 DNA sequences for *S. enteritidis*), we chose Clone SE-3 (60 nt) to be used in our APC-Cas system based on its sequence length, dissociation constant (K_d), bacteria growth suppression and other performances. We have

added the reference and associated description on **page 7** in the revised manuscript.

AP in APC-Cas system is a single-stranded DNA molecule, DNA molecules are susceptible to nuclease-mediated degradation. Unmodified DNA molecules are quickly degraded by nucleases in biological samples, especially in blood.^{2,3} It is reported that 3' exonuclease is primarily responsible for the degradation of oligonucleotides in serum, and phosphorylation at the 3' end of DNA can render the DNA molecule resistant to enzymatic hydrolysis.⁴ Therefore, a phosphate group was labeled at the 3' end of AP as mentioned in the manuscript. In addition, 3' phosphorylation can also prevent DNA polymerase from recognizing DNA ends and then inhibit extension reactions. We have added the references and associated description on **page 5** in the revised manuscript.

1. Kolovskaya, O.S. et al. Development of Bacteriostatic DNA Aptamers for Salmonella. *Journal of Medicinal Chemistry* **56**, 1564-1572 (2013).
2. Beigelman, L. et al. Synthesis and biological activities of a phosphorodithioate analog of 2',5'-oligoadenylate. *Nucleic acids research* **23**, 3989-3994 (1995).
3. Wang, R., Wu, H., Niu, Y. & Cai, J. Improving the stability of aptamers by chemical modification. *Current medicinal chemistry* **18**, 4126-4138 (2011).
4. Gamper, H. et al. Facile preparation of nuclease resistant 3' modified oligodeoxynucleotides. *Nucleic acids research* **21**, 145-150 (1993).

Reviewer #3 (Remarks to the Author):

Manuscript Review:

Title: Single Pathogen Detection via Allosteric Probe-initiated Catalysis and CRISPR-Cas13a Amplification Reaction"

Authors: Shen et al.

1- The manuscript by Shen et al describes a new method for microbial detection. The method is based on the combination of two known techniques: allosteric DNA probes and CRISPR-Cas enzymes. The Allosteric DNA probes have already been used for detection, and the authors here combine these probes with a CRISPR-Cas system with the purpose of adding

two levels of signal amplification through DNA replication and transcription to improve the detection limit. This combination is novel and has not been reported before, and to my knowledge, this is the first use of CRISPR techniques for microbial detection. The manuscript is well-written and technically sound.

2- I have two main concerns:

A- The problem with aptamers: The efficiency of using aptamers for large targets such as microorganisms. Despite the abundant literature in the field, there is no commercial detection assay using aptamers as receptors because of their lack of specificity and poor reproducibility. The only commercial use of aptamers today is for therapeutic reasons and on very small targets (molecules). Given these concerns, the authors should at least provide more information on the aptamers used in this work. Are they commercially obtained, are they produced by the team? did they characterize the binding properties of the aptamers (affinity, dissociation constants)?

Answer: We thank the reviewer for this comment. Reviewer #1 and #2 also raised the similar concern. The DNA sequence of aptamer domain in allosteric probe (AP) used in this assay was discovered by Kolovskaya et al.¹ Kolovskaya et al. has developed a Cell-SELEX technique for selecting highly specific DNA aptamers to *S. enteritidis*. We chose Clone SE-3 (60nt) out of the aptamer pool SE7 (23 DNA sequences for *S. enteritidis*) to be used in our APC-Cas system based on its sequence length, dissociation constant (K_d), bacteria growth suppression and other performances. The APs are commercially obtained from Sangon Biotechnology Co. Ltd (Shanghai, China). The dissociation constant of the chosen aptamer is 7.8 ± 6.1 nM. We have added the reference and associated description on **page 7** in the revised manuscript.

1. Kolovskaya, O.S. et al. Development of Bacteriostatic DNA Aptamers for Salmonella. *Journal of Medicinal Chemistry* **56**, 1564-1572 (2013).

B-The problem with “single pathogen” detection: One of the major problems when we attempt to perform low detection limits with low number of cells is that the probability of having microorganisms in your diluted samples decreases exponentially. If we dilute 10 times a 1mL solution with 10 CFU, the result is rarely 1 CFU per mL. The chance to get 1 CFU per

mL is very slim. As a result, one should analyze all the diluted samples to ensure a positive detection, and even then, it's hard to demonstrate that there is only one single pathogen your sample. I would suggest avoiding titles with "single pathogen" detection claims.

Answer: We agree with the reviewer's comment that limited dilution method is hard to guarantee there is only one single pathogen in the gradually diluted sample. As mentioned in our original manuscript on **Page 12**, "To verify the sensitivity of APC-Cas, 1, 3 and 8 *S. enteritidis* cells were picked out and performed with APC-Cas, individually." We used

Supplementary Fig. 3 | The amount of *S. enteritidis* were quantified by flat colony counting method.

micropipette to pick out 1, 3, 8 *S. enteritidis* cells to perform the experiments, and the picked amount of *S. enteritidis* was further quantified and demonstrated by the plate colony counting method (**Supplementary Fig. 3**). Therefore, we think single pathogen detection could be claimed in the title of our manuscript.

3- Cost of the assay: One of the main motivations of the authors is to offer a sensitive and affordable microbial detection assay. It is true that the developed assay overcomes the need for antibodies. However, this assay uses three different types of enzymes: DNA polymerase, RNA polymerase and CRISPR-Cas13a, and require equipment for DNA amplification, which would significantly increase the cost of the assay. It would be helpful if the authors could estimate the cost of the assay for a single test.

Answer: Thank you for this suggestion. Although APC-Cas system contains three different types of enzymes: DNA polymerase, RNA polymerase and CRISPR-Cas13a, the total reaction volume is 10 μ L and the tested sample volume is only 2.5 μ L. Therefore, the amount of enzymes and other components used in this assay is quite small for a single test. As shown in **Supplementary Table 5**, the cost of APC-Cas system for 1000 reactions is \$858.08, so the cost of the assay for a single test can be as low as \$0.86. We have added **Supplementary Table 5** and associated description on **page 22** in the revised manuscript.

Supplementary Table 5 | APC-Cas cost analysis

Reaction of APC-Cas	Component	Amount	Vendor	Cost (\$)	Fraction used/reaction	Cost/1000 reactions (\$)
Primary amplification	AP	2 OD	Sangon Biotech	67.63	4.54E-04	30.7162
	primer	2 OD	Sangon Biotech	7.05	7.50E-05	0.528375
	dNTP	250 μ L	Takara	11.27	8.00E-04	9.0176
	KF	200 U	NEB	72.14	4.00E-03	288.5632
	RNase-free Water	500 mL	Sangon Biotech	7.50	1.00E-05	0.074959
Secondary amplification	rNTP	500 μ L	NEB	10.93	1.00E-02	109.3384
	T7 RNA Polymerase	2500 U	NEB	314.35	4.00E-04	267.71
	Recombinant RNase Inhibitor	500 U	Takara	13.39	2.00E-02	125.7392
Tertiary amplification	Cas13a	20 nmol	custom purification	95.11	5.00E-06	0.475538
	RNase-free Water	500 mL	Sangon Biotech	7.50	1.00E-05	0.074959
	Reaction buffer	500 μ L	Takara	2.11	1.00E-03	2.1135
	crRNA	100 μ L	custom transcription	10.38	5.00E-05	0.519179
	Reporter probe	2 OD	Takara	276.16	8.40E-05	23.20623
					Total cost/ 1000 reactions (\$)	858.0773

4- Time of detection: The proposed assay involves 5 biochemical steps: target recognition by the aptamers, DNA replication, RNA transcription, RNA recognition by the enzyme Cas13, and Cleavage of the reporter RNA by Cas13. It would be helpful if the authors estimate the

time needed for each step and the overall time of the assay.

Answer: We thank the reviewer for this suggestion. The reaction time of each step has been mentioned in the “Methods” section of manuscript on **page 28**. The time of target recognition by the aptamers, DNA replication and RNA transcription is 30, 20, 60-min, respectively. The RNA recognition by the Cas13a/crRNA complex and the cleavage of reporter probe by Cas13a are carried out in the same reaction system and are completed in one step within 30 min. The overall time of this assay is 140 min.

REVIEWERS' COMMENTS:

Reviewer #1 (Remarks to the Author):

This study is well-designed and the conclusions appear to be supported by the data. As I mentioned in my initial review, I believe that this study is innovative and original.

The authors have revised the manuscript substantially and have addressed many of my original questions and concerns. Details of the aptamer sequence and binding affinity have been added. The authors have expanded the false positive testing to include other bacteria, including several gram-negative samples. These and other additions have strengthened the conclusions of the study.

Reviewer #3 (Remarks to the Author):

[No further comments for author.]

Our response to each review comment is given below (marked in blue).

REVIEWERS' COMMENTS:

Reviewer #1 (Remarks to the Author):

This study is well-designed and the conclusions appear to be supported by the data. As I mentioned in my initial review, I believe that this study is innovative and original.

The authors have revised the manuscript substantially and have addressed many of my original questions and concerns. Details of the aptamer sequence and binding affinity have been added. The authors have expanded the false positive testing to include other bacteria, including several gram-negative samples. These and other additions have strengthened the conclusions of the study.

Answer: We like to thank this reviewer for his/her kind approval.

Reviewer #3 (Remarks to the Author):

[No further comments for author.]

Answer: We like to thank this reviewer for his/her kind approval.